# The Role of Physical Activity Status in the Relationship between Obesity and Carotid Intima-Media Thickness (CIMT) in Urban South African Teachers: The SABPA Study

**DOI:** 10.3390/ijerph19106348

**Published:** 2022-05-23

**Authors:** Tamrin Veldsman, Mariette Swanepoel, Makama Andries Monyeki, Johanna Susanna Brits, Leoné Malan

**Affiliations:** 1Physical Activity, Sport and Recreation Research Focus Area (PhASRec), Faculty of Health Sciences, North-West University, Private Bag X6001, Potchefstroom 2520, South Africa; 12262404@nwu.ac.za (M.S.); andries.monyeki@nwu.ac.za (M.A.M.); 2Hypertension in Africa Research Team (HART), North-West University, Private Bag X6001, Potchefstroom 2520, South Africa; 10069917@nwu.ac.za; 3Technology Transfer and Innovation-Support Office, North-West University, Private Bag X1290, Potchefstroom 2520, South Africa; leonemalan6@gmail.com

**Keywords:** atherosclerosis, carotid intima-media thickness, obesity, physical activity, South Africa, teachers

## Abstract

Globally, the prevalence of physical inactivity and obesity are on the rise, which may increase carotid intima-media thickness (CIMT) as a marker of subclinical atherosclerosis. This study assessed the association between physical activity (PA), obesity, and CIMT. A cross-sectional study design was used, including a sub-sample (*n* = 216) of teachers who participated in the Sympathetic Activity and Ambulatory Blood Pressure in Africans (SAPBA) study. Measurements included the following: physical activity status (measured with ActiHeart devices over 7 consecutive days), body mass index (BMI), waist circumference (WC), waist-to-height ratio (WtHR), CIMT (measured by SonoSite Micromax ultrasound), blood pressure (BP), fasting C-reactive protein (CRP), and cholesterol and glucose levels. Data were analysed using the Statistical Package for Social Science. One-third of the teachers were physically inactive (33%) and had low-grade inflammation CRP ≥ 3 mg/L (41%). Males were more sedentary and had higher BP and CIMT (*p* < 0.05). Independent of age and sex, WC or central obesity was 2.63 times more likely (*p* = 0.02) to contribute to atherosclerosis, especially in females (OR: 4.23, *p* = 0.04). PA levels were insignificantly and negatively (β −0.034; 0.888; 0.240) related to subclinical atherosclerosis. The cardiovascular disease risk profiles and limited PA status may have curbed the beneficial impact of PA on the obesity and atherosclerosis.

## 1. Introduction

Physical inactivity is a significant public health concern [1] and a leading risk factor in the development of various non-communicable diseases (NCDs) [2]. Globally, one in four adults is physically inactive [2,3] and the prevalence of physical inactivity is increasing in many developed and developing countries [4] with one-third of the South African adult population being classified as sedentary [3]. Teachers, in particular, spend most of their working day in sedentary or light energy-cost activities [5].

This increasing prevalence in physical inactivity may pose significant implications for the prevalence of NCDs such as cardiovascular disease (CVD), diabetes, and cancer, as well as for modifiable risk factors such as elevated BP, raised glucose, and increased prevalence of overweight and obesity [4]. Between 1980 and 2008, the prevalence of global obesity, as measured by body mass index (BMI), nearly doubled [6]. Of the South African adult population, 51.9% were overweight (BMI ≥ 25 kg/m^2^) [7] and 27% were obese (BMI ≥ 30 kg/m^2^) [8]. Among South African teachers in Cape Town, the prevalence of overweight was 31.1%, 53.6% were obese, and 18.7% of the teachers are at high risk of having a heart attack or stroke in the next decade [9].

Cardiovascular disease is a major cause of death in the general population [10]. Early detection of CVD occurrences before full manifestation can be assessed by measuring carotid intima-media thickness (CIMT) [11]. Carotid intima-media thickness is a non-invasive subclinical measure to evaluate the progression of atherosclerosis and intima-media thickening [11,12]. Obesity may worsen the progression of CIMT and, in particular, visceral/central obesity is a predictor of endothelial dysfunction and is correlated with an increased CIMT in obese individuals [13,14,15,16,17,18].

In a study conducted by Lo et al. [15] on young, healthy American females, the obese females were reported to have a higher CIMT than females who were considered to be overweight or who had a healthy BMI (<25 kg/m^2^), and subcutaneous abdominal fat was more associated with increased CIMT than visceral adiposity. In middle-aged Finnish males with no known atherosclerotic disease, central obesity, as measured by waist-to-hip ratio and waist circumference (WC), was associated with accelerated progression of carotid atherosclerosis [13]. In a healthy middle-aged (<60 years) Chinese population, BMI was associated with an increased CIMT [19]. Therefore, increased abdominal fat is positively associated with an increased CIMT and an increased risk of developing atherosclerosis, increasing the risk of developing cardiovascular disease.

Participation in regular PA influences CIMT, where exercise can prevent the development of atherosclerosis through weight reduction, glucose regulation, fat loss, improved endothelial function, improved glucose sensitivity, and increased cardiorespiratory fitness, leading to controlled lipid values and lowered BP values [20,21]. Higher levels of PA and decreased sedentary time are associated with a lowered risk of developing a CVD, as indicated by CIMT [22]. In a study of Korean office workers, a significant positive relationship was found between central obesity and an increased CIMT among the physically inactive adults, but the relationship was not found in the office workers who were considered to be physically active [10].

Little is known about the role of PA in the relationship between obesity and CIMT, especially among South African teachers. Due to the nature of the occupation of teaching, levels of inactivity in teachers are high and there is a high prevalence of obesity, factors that place a burden on the South African education system [9]. Thus, this study focused on the association between PA, obesity, and CIMT in a multi-ethnic cohort of South African teachers. It was hypothesised that CIMT would be significantly positively associated with WC, and physical activity would be negatively associated with central obesity and atherosclerosis.

## 2. Materials and Methods

### 2.1. Study Design

This cross-sectional study was part of the Sympathetic Activity and Ambulatory Blood Pressure in Africans (SABPA) prospective cohort study. The baseline measurements for the SABPA study commenced in 2008/2009, and follow-up data were collected similarly to the baseline data in 2011/2012. The detailed methodology of the SABPA study has been published elsewhere and the description of the method partly reproduces its wording [23]. The Health Research Ethics Committee (HREC) of North-West University (Potchefstroom campus: NWU-0036-07-S6) approved the SABPA study, which adhered to the Declaration of Helsinki (2014). The SABPA study gained consent and cooperation from the North West Province Department of Education, the South African Demographic Teachers Union, and the principals of the schools. All participants voluntarily signed an informed consent form before data were collected.

### 2.2. Study Population and Sample

All urban-dwelling South African school teachers enrolled in the 43 schools in the Dr. Kenneth Kaunda Education District residing in the North West Province of South Africa were invited to participate in the SABPA study (*n* = 2170) (Figure 1). The participants were of similar socioeconomic backgrounds and were screened to meet eligibility criteria. Pregnant and lactating participants were excluded from the study, as well as participants using alpha- and beta-blockers or psychotropic substances, those who had donated blood or were vaccinated in the past 3 months, and those with tympanum temperature of ≥37.5 °C. The exclusion followed in our study is grounded on the endocrine pathways associated with lactation, which is thought to be linked to factors that influence systolic blood pressure such as arterial stiffness and compliance [24,25]. After screening, 409 participants were eligible for data collection. The participants were from urban-dwelling well-educated Black (African) and White African (Caucasian) male and female teachers with similar socioeconomic standing [23]. The secondary data of participants who completed the follow-up measurements and who wore the ActiHeart device for a full seven consecutive days or had less than 40 min of daily non-contact time during awake hours (*n* = 216) were included in this sub-study (Figure 1).

### 2.3. Anthropometric Measures

The anthropometric measurements of participants were taken by two level 2 kinanthropometrists according to the methods of the International Society for the Advancement of Kinanthropometry (ISAK) [26]. Height (cm), weight (kg), and WC (cm) were measured. The height and weight of participants were used to calculate BMI (kg/m^2^) and WHtR. Participants’ WCs were classified according to the ethnic-specific cut-off points: black males WC ≥ 90 cm; black females ≥ 98 cm [27]; Caucasians JIS [28] > 94 cm in males, ≥80 cm in females. A WHtR of ≥0.5 was used to indicate an increased risk of CVD [29]. BMI was classified according to the cut-off points of the American College of Sports Medicine (ACSM) as follows: underweight BMI < 18.5 kg/m^2^; normal weight BMI between 18.5 kg/m^2^ and 24.9 kg/m^2^; overweight BMI between 25.0 kg/m^2^ and 29.9 kg/m^2^; and obesity BMI ≥ 30 kg/m^2^ [30]. Intra- and inter-observer variability was less than 5%.

### 2.4. Objectively Measured Physical Activity

Participants’ weekly habitual physical activity (PA) was objectively measured over seven consecutive days with a combined heart rate and accelerometer—the ActiHeart (GB06/67703; CamNtech Ltd., Upper Pendrill Court, Papworth Everard, Cambridgeshire, CB233UY, UK). The resting heart rate of the participants was obtained by a registered nurse using a 12-lead electrocardiogram (NORAV Medical Ltd. PC 1200, software v.5.030, Kiryat Bialik, Israel). The resting heart rate was used to calculate the sleeping heart rate (resting heart rate minus 10 beats per minute) as required to program the ActiHeart device. Individual step test calibrations were not performed due to time constraints, the vast amount of data to be collected, and the CVD risk profile of the participants; however, a biokineticist (clinical exercise physiologist) thoroughly questioned each participant about their daily and weekly PA patterns before an activity level was selected on the ActiHeart program. Each participant’s ActiHeart data were visually inspected to distinguish between sleep- and awake-time. Sleep- and awake-time of the participants were established using heart rate, metabolic equivalent of task (MET), and activity levels. Sleep-time was defined as a period where there was a gradual decrease in heart rate, activity levels at zero during the evenings, and where the heart rate gradually decreased (for 15 or more epochs) in the evenings to less than the average heart rate in a selected awake-time sedentary sample period [31]. The epoch for the full 7 day collection was set at 60 s intervals.

Awake-time was indicated by an immediate increase in heart rate of more than 10–20 beats per minute, as well as increased MET and an increase in activity level. The ActiHeart software was used to derive daily time spent in various MET categories. The different MET categories were grouped according to time spent sedentary (≤1.5 MET) and time spent participating in light-intensity PA (1.5–3 MET) [5]. Activity energy expenditure (AEE), total energy expenditure (TEE), and PA level (PAL) were determined by the ActiHeart using inbuilt equations based on a branched model approach [32]. The PAL was calculated as TEE/resting energy expenditure (REE) [33]. After data analysis, only two PA groups, namely sedentary (≤1.5 MET) and light-intensity PA (1.5–3 MET) could be identified. Only one participant was classified as participating in moderate-to-high PA (>3 MET) and based on statistical power principles this individual was excluded from the analyses.

### 2.5. Subclinical Atherosclerosis

The CIMT of participants was obtained using a SonoSite Micromaxx ultrasound system (SonoSite Inc., Bothell, WA, USA) and a 6–13 MHz linear array transducer, using previously described protocols [11]. Images of at least two optimal angles of the left and right common carotid artery were obtained. The images were digitised and imported to Artery Measurement Systems automated software (AMS, Gothenburg, Sweden, v1.130) for analysing CIMT [34,35]. A maximal 10 cm segment with good quality imaging was used for analysis. The program automatically detects the borders of the intima-media of the near and far wall, as well as the inner diameter of the vessel, and calculates CIMT from around 100 discrete measurements through the 10 cm segment. This automated analysis was capable of being manually corrected if not appropriate on visual inspection. For this study, the far-wall left CIMT measurements were used. Intra-observer variability was 0.04 mm between two measurements taken 4 weeks apart on the same 10 participants. CIMT of ≥0.75 mm was considered indicative of moderate atherosclerotic risk and ≥0.9 mm was regarded as high risk [36]. The images were also examined for the presence of plaque at the right and left bifurcation of the internal carotid artery. Plaque was defined as a focal structure encroaching into the arterial lumen by at least 0.5 mm or by 50% of the surrounding intima-media thickness or demonstrating a thickness >1.5 mm [11].

#### 2.5.1. Blood Pressure

The resting BP of the participants was measured with a sphygmomanometer (1.3M^TM^ Littman^®^ II S.E. Stethoscope 2205, Riesster CE 0124, No. 1010–108 Diplomat-presameter^®^, Jungingen, Germany) on the non-dominant arm with an appropriate cuff size using the Rica/Rocci Korotkoff method. The measurements were repeated after 5 min and the second measurement was used for the metabolic syndrome criteria for BP [28]. The metabolic syndrome criteria for elevated blood pressure values are ≥130/85 mmHg or if the participant uses antihypertensive treatment [28].

#### 2.5.2. Biochemical Analysis

A registered nurse obtained fasting resting blood samples with a winged infusion set from the brachial vein branches of the dominant arm before 9:00 a.m. All blood samples were obtained from never-thawed serum/plasma/citrate samples. Glucose was collected in sodium fluoride tubes, and metabolic syndrome markers were handled according to standardised procedures and stored at −80 °C [23]. The following International Diabetes Federation (IDF) cut-points for blood analyses were used: high-density lipoprotein (HDL) ≤ 1.03 mmol/L, triglycerides ≥ 1.70 mmol/L, and fasting glucose ≥ 5.60 mmol/L [28]. These values and ultra-sensitive C-reactive protein (CRP) levels were determined with an enzyme rated method (Unicel DXC 800—Beckman and Coulter, Munich, Germany); homogeneous immunoassay (Modular ROCHE Automized systems, Basel, Switzerland); and a particle enhanced turbidimetric assay (Cobas Integra 400 plus, Roche, Basel, Switzerland), respectively. CRP ≥ 3 mg/L or low-grade inflammation was regarded as a high risk for CVD [37]. Inter-and intra-variability was <5%. Liver enzyme, serum gamma-glutamyl transferase (GGT; U/L), as an indicator of alcohol use was measured using enzymatic colorimetric assay, Cobas Integra 400 plus (Roche, Basel, Switzerland). Inter-and intra-variability were <5% for all analyses.

### 2.6. Data Analysis

Statistical analyses were performed using SPSS v.26 (SPSS Inc., Chicago, IL, USA). Normality of the data was achieved using the Shapiro–Wilk test and quantile–quantile plots; variables that were not normally distributed were log-transformed. Descriptive analyses (means, standard deviations, and frequencies) were performed for all PA, anthropometric, CIMT measurements, and risk factors for CVD. The independent *t*-test for normally distributed data and chi-squared test for categorical variables were performed to determine significant differences by sex, and sedentary and light PA groups. A Spearman’s rho correlation matrix was calculated for anthropometric measurements, AEE, and CIMT between the sedentary and light PA groups. A binomial logistic regression analysis was performed to assess the probability of central adiposity to predict CIMT (≥0.75 mm) in a teachers’ cohort. CIMT was the dependent variable with age, sex, GGT, SBP or DBP, CRP, ethnicity-specific WC, and PA included as independent variables. A significant relationship was found between CIMT and WC for the total group, and then data were analysed separately for males and females. Correlation coefficient values were classified as follows: <0.10 = a weak correlation, values of 0.30–0.50 indicate a moderate correlation, and ≥0.05 indicates a strong correlation. The significance level was set at *p* ≤ 0.05 [38].

## 3. Results

Table 1 displays the basic lifestyle, anthropometric, and BP characteristics of the study population according to sex. Male teachers were significantly taller (*p* < 0.001) and heavier (*p* < 0.001) than female teachers. The findings also indicated that male teachers had a significantly greater WC than female teachers (102.12 ± 12.79 cm vs. 91.08 ± 16.74 cm; *p* < 0.001). Male teachers also had significantly higher systolic BP (SBP) (135 ± 17 mmHg vs. 124 ± 18 mmHg; *p* < 0.001), diastolic BP (DBP) (91 ± 10 mmHg vs. 82 ± 10 mmHg; *p* < 0.001), and CIMT (0.75 ± 0.16 mm vs. 0.66 ± 0.12 mm; *p* < 0.001) values than female teachers. For the entire group of teachers, 74 were overweight (34%) and 84 were obese (39%) (Figure 2). Male teachers had a higher prevalence of overweight (42%) compared to female teachers (27%). However, more female teachers were obese (41%) than male teachers (37%). Furthermore, male teachers self-reported higher smoking (20%) and alcohol usage (66%) habits than female teachers. The results suggest that 14% of the total participants (20% of male and 8% of female teachers) were smokers. Overall, more than one-third of the teachers were physically inactive (33%) (Figure 3) and had raised CRP ≥ 3 mg/L (41%). More males had a CIMT of ≥0.75 mm than females (*p* < 0.001) (Figure 4).

In Table 2, the participants’ descriptive characteristics are displayed according to the WC classification and level of PA. Independent *t*-tests were used to determine the differences between the WC categories in the specific PA groups. In the sedentary group of teachers, there were significant differences between the body composition variables. Teachers classified as overweight and sedentary had significantly larger BMI (32.55 ± 6.14 kg/m^2^; *p* < 0.001), WC (108.71 ± 13.21 cm; *p* < 0.001), and WHtR (0.63 ± 0.09; *p* < 0.001) values than teachers with a normal WC. Similarly to the sedentary teachers, teachers who participated in light PA with larger WC values had significantly larger BMI (31.14 ± 5.98 kg/m^2^; *p* < 0.001), WC (101.34 ± 11.27; *p* < 0.001), and WHtR (0.60 ± 0.07; *p* < 0.001) values than the participants who had a normal WC.

Table 3 shows the independent *t*-test results for sedentary teachers and teachers who participated in light PA, divided into WHtR categories of greater or equal to and less than 0.5. In the sedentary group, the teachers with a WHtR ≥ 0.5 had significantly higher SBP and DBP values, exceeding hypertension threshold cut points of SBP 140 mmHg and/or DBP 90 mmHg (144 ± 19 mmHg vs. 127 ± 18 mmHg; *p* < 0.001), DBP (93 ± 11 mmHg vs. 87 ± 12 mmHg; *p* = 0.03), and PAL (1.50 ± 0.14 vs. 1.42 ± 0.11; *p* = 0.05) compared with teachers with a WHtR < 0.5. Similarly, the teachers who participated in light PA with a WHtR ≥ 0.5 had significantly higher SBP (136 ± 18 mmHg vs. 123 ±15 mmHg; *p* < 0.001), DBP (90 ± 11 mmHg vs. 82 ± 10 mmHg; *p* < 0.001), and PAL (2.36 ± 0.59 vs. 2.12 ± 0.45; *p* = 0.02) when compared with teachers with a WHtR < 0.5.

Table 4 presents the Spearman’s rho correlation coefficients for the participants in the sedentary and light PA groups. In sedentary participants, a moderate significant positive relationship between age and CIMT (*r* = 0.33; *p* = 0.01) was found. In teachers who participated in light PA, a moderate significant positive relationship was found between age and CIMT (*r* = 0.31; *p* < 0.01). There was also a weak significant positive relationship between CIMT and body weight (*r* = 0.22; *p* = 0.01), and WC (*r* = 0.19; *p* = 0.02).

A logistic regression analysis was performed to ascertain the effects of age, sex, GGT (alcohol consumption), SBP/DBP, CRP, WC, and PA on the likelihood that participants have subclinical atherosclerosis as shown in Table 5. The logistic regression model was statistically insignificant (χ^2^ (8) = 4.288, *p* = 0.830), and it is explained by 20.3% of the variance in subclinical atherosclerosis and correctly classified 68.2% of cases. Independent of age and sex, WC or central obesity was 2.63 times more likely (*p* = 0.02) to contribute to atherosclerosis. Increasing PA levels or CRP was not related to subclinical atherosclerosis, *p* > 0.09 (OR of 0.97; β = −0.034; *p* = 0.888).

When data were analysed by sex, it was found that WC (OR: 4.23, *p* = 0.036) significantly contributed to atherosclerosis in females, whilst age demonstrated a borderline effect in both females and males (Table 6). The overall model for females was statistically significant, χ^2^(6) = 14.64, *p* < 0.05, whilst in males the model was insignificant χ^2^(6) = 4.72, *p* > 0.05. The application of the Cox and Snell R^2^ or Nagelkerke R^2^ methods in the model for females showed that the explained variation in the dependent variable ranges from 13.0% to 19.0%. In males, application of the Cox and Snell R^2^ or Nagelkerke R^2^ in the model showed that the explained variation in the dependent variable ranges from 4.5% to 6.0%.

## 4. Discussion

The purpose of this study was to investigate the relationship between PA, obesity, and subclinical atherosclerosis in teachers in the North West Province of South Africa. The results of the study showed significant positive relationships between CIMT and WC in teachers utilising light PA. WC significantly contributes to atherosclerosis in females compared to males. The current findings show that more than one-third of the teachers were overweight or obese, sedentary, and had elevated CRP values. Male teachers had significantly higher BP, CIMT, CRP, and sedentary levels when compared with female teachers. The results of this study are congruent with the results of Jin and colleagues [10] among Korean office workers.

In accord to our findings of a positive relationship between CIMT and WC, Jin et al. [10] further indicated that physical inactivity in sedentary Korean office workers may lead to a higher risk of central obesity and an elevated CIMT that was not seen in physically active persons. Increasing PA levels were however not related to CIMT. Our findings directly oppose those of Jin et al. [10] where it was indicated that PA was a predictor of abnormally increased CIMT. This may be due to 33% of our participants being sedentary and overweight and the remaining 67% only participating in light-intensity PA, indicating that the intensity of PA may have been too low to influence the relationship between obesity and CIMT. We suggest that teachers might spend most of their working time seated, standing, and walking, which are all low energy-cost activities [5].

Another reason for our findings that PA per se did not predict CIMT, but may have been due to the high prevalence of overweight (34%) and obesity (39%) in this sample population. Increased levels of obesity are related to subclinical disease, which contributes to the risk of CVD [39]. Increased BMI in obese and overweight adolescents revealed a positive association with CIMT, independent of factors such as age, sex, SBP, and lipid profiles [40]. Ryder et al. [41] suggested that obesity, diabetes, and elevated blood pressure in adolescents are linked with early development in vascular ageing [41]. The observed prevalence of overweight and obesity in the current study was consistent with a recent South African study among municipal workers in South Africa [42] and with the South African statistics noted in the recent World Health Organization Overweight and Obesity report [43].

The weak significant positive association between WC and CIMT was only observed among the teachers who were sedentary and participated in light PA. This result is in agreement with previous results whereby strong associations between central obesity and the risk of developing CVD were reported [12,44,45,46] and congruent with information in articles on middle-aged adults from Finland, the USA, Korea, and China [13,15,19]. In our study, we could also not replicate the positive association between CIMT and BMI, which is in contrast to the study of Arnold et al. [19], where a weak positive association existed between CIMT and BMI in middle-aged Chinese adults. BMI levels are, however, not the best risk marker to determine CVD risk as BMI measures per se are flawed and not used to distinguish between fat and lean tissue or consider sex or multi-ethnic differences [47]. WC significantly contributes to atherosclerosis in females, whilst age demonstrated a borderline effect in both females and males in this study. A possible reason for WC significance in females and not males may be due to the high levels of obesity observed in the female sample in the study.

A concerning matter is that more than one-third (41%) of the teachers’ cohort in the present study had elevated CRP ≥ 3 mg/L (low-grade inflammation). Our findings are therefore similar to the EPIC-Norfolk study among middle-aged adults (40–79 years), which revealed that more than a third of their participants had elevated CRP [48]. The higher CRP prevalence (41%) may be explained by the high prevalence of overweight and obesity in our study population, as CRP (an inflammatory marker) is often increased in persons with increased central obesity [6]. Indeed, central obesity (WC) has furthermore been recognised as an inflammatory mediator [49] and was related to metabolic perturbations in the SABPA cohort [50], as well as with chronic stress, vascular dysregulation, and cognitive dysfunction [51,52]. The ophthalmic artery is the first branch of the internal carotid artery and underscores the risk of vascular dysregulation and even stroke [53]. Therefore, the relationship between CIMT and obesity may indicate a decline in well-being and should be considered in preventive strategies for subclinical atherosclerosis. Our study accentuates the importance of assessing cardiometabolic risk in South African teachers regardless of their PA status. Teaching is a physically inactive occupation and future research should focus on providing PA interventions to teachers to reduce cardio-metabolic risk factors, especially physical inactivity, and overweight/obesity.

Potential limitations to this study may include the cross-sectional design of the study, which prohibited the possible detection of CIMT and PA trajectories. Future studies should focus on a longitudinal design to investigate changes in CIMT and PA over time. The study population is not representative of all South African teachers as teachers were only sampled in one province of South Africa and the generalisation of the results is not representative of the whole South African population. The strict adherence to the 7 day wearing time of the ActiHeart device limited the study sample. Another limitation is that individual step test calibration of the ActiHeart device was not performed due to the vast number of participants, limited time, and the high-risk profiles of the participants; however, participants were thoroughly interviewed about their habitual PA patterns. An additional limitation of the study was the lack of menopausal status data given that some studies established that in obese women, significant differences in CIMT between pre- and post-menopausal women, and menopause exist [54,55,56]. The role of genetics factors in the physiological parameters of individuals to health is another limitation and should be considered in future similar studies.

## 5. Conclusions

In line with the study hypothesis, significant positive relationships between CIMT and WC were observed. In addition, although not significantly, physical activity was negatively associated with CIMT and BMI. WC significantly contributes to atherosclerosis in females compared to males. The study reported a high prevalence of physical inactivity, high CRP, and overweight/obesity in the sampled population, and these factors may have diminished the protective role of PA in the relationship between CIMT and obesity. Given the risk profile of the teachers, effective PA intervention and education programs should be implemented to reduce their CVD risk.

## Figures and Tables

**Figure 1 ijerph-19-06348-f001:**
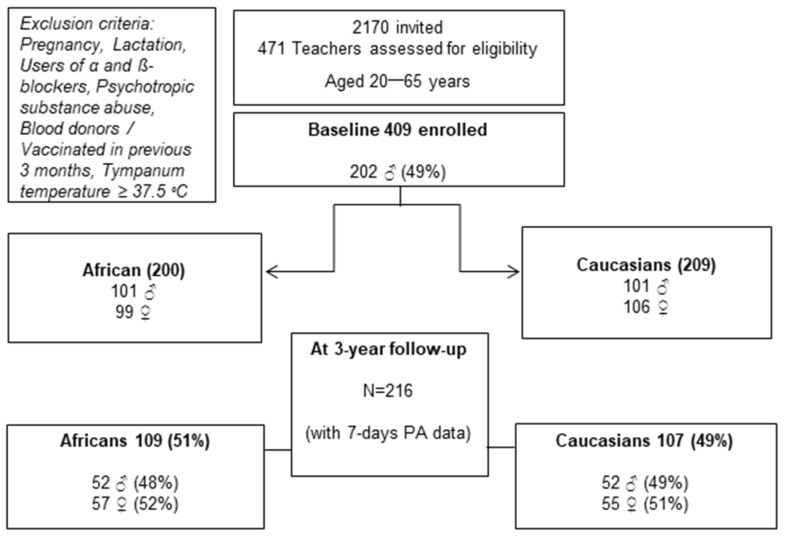
Secondary data drawn from the Sympathetic Activity and Ambulatory Blood Pressure in Africans (SABPA) prospective cohort study population.

**Figure 2 ijerph-19-06348-f002:**
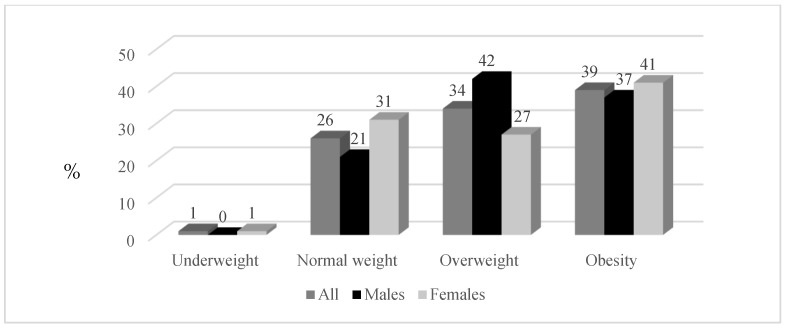
Percentage (%) scores for BMI categories.

**Figure 3 ijerph-19-06348-f003:**
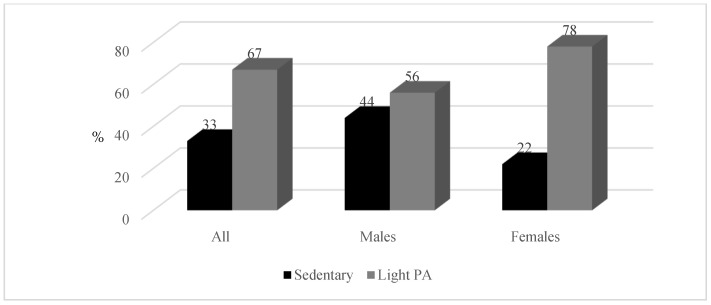
Percentage (%) scores for physical activity.

**Figure 4 ijerph-19-06348-f004:**
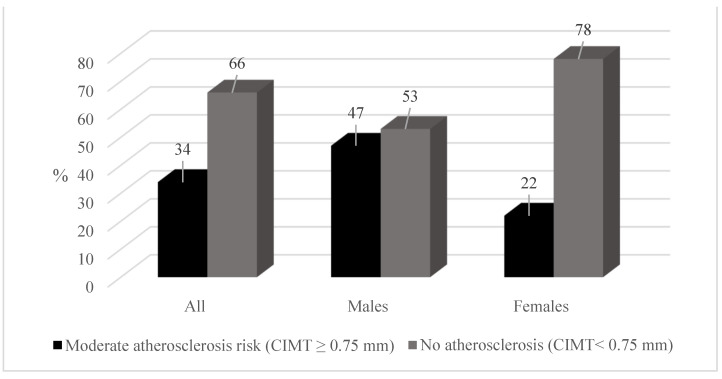
Percentage (%) scores for CIMT.

**Table 1 ijerph-19-06348-t001:** Differences in descriptive characteristics of an ethnic sex cohort.

	Total Group (*n* = 216)	Males (*n* = 104)	Females (*n* = 112)
Age (years)	49.67 ± 8.44	49.79 ± 8.48	49.55 ± 8.43
Height (cm)	169.12 ± 10.18	176.6 ±7.90 *	162.14 ± 6.37 *
Weight (kg)	83.65 ± 19.62	90.75 ± 16.51 *	77.06 ± 20.04 *
BMI (kg/m^2^)	29.3 ±6.49	29.06 ± 4.77	29.64 ± 7.76
WC (cm)	96.40 ± 15.92	102.12 ± 12.79 *	91.08 ± 16.74 *
WHtR	0.57 ± 0.09	0.58 ± 0.07	0.56 ± 0.11
SBP (mmHg)	129.11 ± 18.29	134.57 ± 16.56 *	124.04 ± 18.42 *
DBP (mmHg)	86.22 ± 11.33	90.87 ± 10.42 *	81.91 ± 10.44 *
CIMT (mm)	0.70 ± 0.15	0.75 ± 0.16 *	0.66 ± 0.12 *
Hypertension medication, *n* (%)	67 (31)	39 (38)	28 (25)
Metabolic syndrome, *n* (%)	63 (29)	48 (46) *	15 (13)
Physical activity			
Sedentary, *n* (%)	71 (33)	46 (44)	25 (22) *
Light PA, *n* (%)	145 (67)	58 (56)	87 (78)
CRP ≥ 3 mg/L, *n* (%)	88 (41)	36 (35)	52 (47)
Moderate atherosclerosis risk (CIMT ≥ 0.75 mm)	74 (34)	49 (47) *	25 (22) *
No atherosclerosis risk (CIMT < 0.75 mm)	141 (66)	55 (53)	86 (78)

The *t*-test values for independent groups are displayed as arithmetic mean ± standard deviation. Chi-squared (χ^2^) tests were used to determine proportions and prevalence and are indicated as frequencies (%). * Level of significance was set at *p* ≤ 0.05; BMI = body mass index; CIMT = carotid intima-media thickness; CRP = C-reactive protein; DBP = diastolic blood pressure; SBP = systolic blood pressure; WC = waist circumference; WHtR = waist-to-height ratio.

**Table 2 ijerph-19-06348-t002:** Differences in descriptive characteristics of the teachers according to waist circumference classification.

	WC Classification	Sedentary		Light PA	
		*n*	Mean ± SD	*n*	Mean ± SD
Age (years)	Overweight	48	49.60 ± 9.08	105	50.85 ± 8.17
	Normal WC	23	46.91 ± 9.76	40	48.23 ± 7.14
Height (cm)	Overweight	48	172.82 ± 9.60 *	105	169.98 ± 10.40 *
	Normal WC	23	164.24 ± 9.49	40	165.25 ± 8.55
Weight (kg)	Overweight	48	96.96 ± 17.06 *	105	88.78 ± 16.68 *
	Normal WC	23	63.00 ± 9.70	40	66.09 ± 10.93
BMI (kg/m^2^)	Overweight	48	32.55 ± 6.14 *	105	31.14 ± 5.98 *
	Normal WC	23	23.38 ± 3.36	40	24.32 ± 4.10
WC (cm)	Overweight	48	108.71 ± 13.21 *	105	101.34 ± 11.27 *
	Normal WC	23	79.72 ± 7.28	40	78.23 ± 7.99
WHtR	Overweight	48	0.63 ± 0.09 *	105	0.60 ± 0.07 *
	Normal WC	23	0.49 ± 0.05	40	0.47 ± 0.05
CIMT (mm)	Overweight	47	0.72 ± 0.17	105	0.73 ± 0.15 *
	Normal WC	23	0.65 ± 0.15	40	0.66 ± 0.10
AEE (kcal/wk)	Overweight	48	704.09 ± 636.28	105	1526.54 ± 1179.74
	Normal WC	23	990.43 ± 858.24	40	1346.84 ± 965.68
TEE (kcal/wk)	Overweight	48	2807.59 ± 701.14	105	3571.55 ± 1431.79
	Normal WC	23	2830.08 ± 1100.43	40	3247.30 ± 1589.70
PAL	Overweight	48	1.48 ± 0.15	105	2.37 ± 0.66
	Normal WC	23	1.51 ± 0.13	40	2.26 ± 0.50

* Level of significance was set at *p* ≤ 0.05; sedentary = ≤1.5 metabolic equivalent of task (MET); light physical activity (PA) = 1.5–3 MET; WC ethnic specific cut-off points: black males WC = ≥90 cm; black females = ≥98 cm [25]; Caucasians JIS [26]: >94 cm males, ≥80 cm females; AEE = activity energy expenditure; BMI = body mass index; CIMT = carotid intima-media thickness; DBP = diastolic blood pressure; PAL = physical activity level; SBP = systolic blood pressure; SD = standard deviation; TEE = total energy expenditure; WC = waist circumference; WHtR = waist-to-height ratio.

**Table 3 ijerph-19-06348-t003:** Independent *t*-test for participants according to waist-to-height ratio category.

	WHtR Category	Sedentary		Light PA	
		*n*	Mean ± SD	*n*	Mean ± SD
Age (years)	<0.5	44	47.32 ± 10.21	95	49.96 ± 7.56
	≥0.5	27	51.04 ± 7.27	49	50.67 ± 8.68
Height (cm)	<0.5	44	170.17 ± 11.33	95	169.79 ± 10.16
	≥0.5	27	169.83 ± 8.64	49	166.60 ± 9.88
Weight (kg)	<0.5	44	75.27 ± 17.77 *	95	75.64 ± 13.17 *
	≥0.5	27	103.37 ± 16.29 *	49	96.36 ± 19.23 *
BMI (kg/m^2^)	<0.5	44	25.67 ± 3.77 *	95	26.24 ± 3.49 *
	≥0.5	27	35.96 ± 6.01 *	49	35.26 ± 6.28 *
WC (cm)	<0.5	44	89.08 ± 12.08 *	95	87.80 ± 10.39 *
	≥0.5	27	116.02 ± 12.42 *	49	109.49 ± 10.07 *
WHtR	<0.5	44	0.52 ± 0.05 *	95	0.52 ± 0.05 *
	≥0.5	27	0.68 ± 0.08 *	49	0.66 ± 0.06 *
SBP (mmHg)	<0.5	44	127 ± 18 *	95	123 ± 15 *
	≥0.5	27	144 ± 19 *	49	136 ± 18 *
DBP (mmHg)	<0.5	44	87 ± 12 *	95	82 ± 10 *
	≥0.5	27	93 ± 11 *	49	90 ± 11 *
CIMT (mm)	<0.5	44	0.70 ± 0.17	95	0.71 ± 0.14
	≥0.5	27	0.69 ± 0.16	49	0.71 ± 0.16
AEE (kcal/wk)	<0.5	12	819.09 ± 970.19	28	1238.22 ± 153.10
	≥0.5	59	792.33 ± 671.42	116	1485.69 ± 98.92
TEE (kcal/wk)	<0.5	12	2578.99 ± 1292.64	28	3085.28 ± 304.02
	≥0.5	59	2862.85 ± 725.99	116	3520.39 ± 121.17
PAL (kcal/wk)	<0.5	12	1.42 ± 0.11 *	28	2.12 ± 0.45 *
	≥0.5	59	1.50 ± 0.14 *	116	2.36 ± 0.59 *

* Level of significance was set at *p* ≤ 0.05; sedentary = ≤1.5 metabolic equivalent of task (MET); light physical activity (PA) = 1.5–3 MET; AEE = activity energy expenditure; BMI = body mass index; CIMT = carotid intima-media thickness; DBP = diastolic blood pressure; PAL = physical activity level; SBP = systolic blood pressure; TEE = total energy expenditure; WC = waist circumference; WHtR = waist-to-height ratio.

**Table 4 ijerph-19-06348-t004:** Spearman’s rho correlation matrix for anthropometric measurements, activity energy expenditure, and carotid intima-media thickness among sedentary and light-intensity physical activity groups.

		Sedentary							Light PA						
		Age (years)	Height (cm)	Weight (kg)	BMI (kg/m^2^)	CIMT (mm)	WC (cm)	AEE (kcal/wk)	Age (years)	Height (cm)	Weight (kg)	BMI (kg/m^2^)	CIMT (mm)	WC (cm)	AEE (kcal/wk)
Age (years)	*r*	–	0.06	0.07	0.03	0.33 **	0.14	−0.13	–	0.02	0.03	−0.01	0.31 **	0.13	0.01
	*p*		0.60	0.59	0.79	0.01	0.25	0.29		0.81	0.68	0.90	<0.01	0.11	0.90
Height (cm)	*r*	0.06	–	0.54 **	0.12	0.39 **	0.37 **	−0.23	0.02	–	0.42 **	−0.16	0.22 **	0.24 **	0.28 **
	*p*	0.60		<0.01	0.31	<0.01	<0.01	0.06	0.81		<0.01	0.06	0.01	<0.01	<0.01
Weight (kg)	*r*	0.07	0.54 **	–	0.88 **	0.14	0.92 **	−0.09	0.03	0.42 **	–	0.77 **	0.22 **	0.86 **	0.21 *
	*p*	0.59	<0.01		<0.01	0.25	<0.01	0.44	0.68	<0.01		<0.01	0.01	<0.01	0.01
BMI (kg/ m^2^)	*r*	0.03	0.12	0.88 **	–	−0.05	0.90 **	0.02	−0.01	−0.16	0.77 **	–	0.09	0.81 **	−0.01
	*p*	0.79	0.31	<0.01	.	0.66	<0.01	0.86	0.90	0.06	<0.01		0.27	<0.01	0.88
CIMT (mm)	*r*	0.33 **	0.39 **	0.14	−0.05	–	0.14	−0.17	0.31 **	0.22 **	0.22 **	0.09	–	0.19 *	0.11
	*p*	0.01	<0.01	0.25	0.66		0.25	0.15	<0.01	0.01	0.01	0.27		0.02	0.20
WC (cm)	*r*	0.14	0.37 **	0.92 **	0.90 **	0.14	–	−0.02	0.13	0.24 **	0.86 **	0.81 **	0.19 *	–	0.10
	*p*	0.25	<0.01	<0.01	<0.01	0.25		0.86	0.11	<0.01	<0.01	<0.01	0.02		0.234
AEE (kcal/wk)	*r*	−0.13	−0.23	−0.09	0.02	−0.17	−0.02	–	0.01	0.28 **	0.21 *	−0.01	0.11	0.10	-
	*p*	0.29	0.06	0.44	0.86	0.15	0.86		0.90	<0.01	0.01	0.88	0.20	0.24	

* Level of significance was set at *p* ≤ 0.05; ** correlation was significant at the 0.01 level (two-tailed); sedentary = ≤1.5 metabolic equivalent of task (MET), light activity = 1.5–3 MET, moderate-to-high activity = ≥3 MET; WC ethnic specific cut-off points: black males WC = ≥90 cm; black females = ≥98 cm [25]; Caucasians JIS [26]: >94 cm males, ≥80 cm females; AEE = activity energy expenditure; BMI = body mass index; CIMT = carotid intima-media thickness; PA = physical activity; WC = waist circumference; WHtR = waist-to-height ratio.

**Table 5 ijerph-19-06348-t005:** Logistic regression analysis to assess the probability of central adiposity to predict subclinical atherosclerosis in a teachers’ cohort.

CIMT ≥ 0.75 mm								
	*β*	S.E.	Wald	df	Sig.	Exp(B)	95% C.I. for EXP(B)	
							Lower	Upper
Age (years)	−0.051	0.020	6.396	1	0.011	0.950	0.913	0.989
Sex	0.853	0.350	5.946	1	0.015	2.346	1.182	4.656
GGT (U/L)	−0.002	0.003	0.303	1	0.582	0.998	0.993	1.004
SBP (mmHg)	−0.008	0.010	0.719	1	0.397	0.992	0.974	1.011
Log CRP	0.036	0.034	1.086	1	0.297	1.036	0.969	1.109
WC cut point	0.966	0.412	5.500	1	0.019	2.628	1.172	5.892
PAL	−0.034	0.240	0.020	1	0.888	0.967	0.604	1.548
Constant	2.663	1.743	2.335	1	0.126	14.346		

Variable(s) entered on step 1: age, sex, GGT, SBP, log CRP, WC cut point, PAL. CRP = C-reactive protein; GGT = gamma glutamyl transferase; PAL = physical activity level; SBP = systolic blood pressure; WC = ethnic specific waist circumference.

**Table 6 ijerph-19-06348-t006:** Logistic regression analysis to assess the probability of central adiposity to predict subclinical atherosclerosis in a male and female teachers’ cohort.

	B	S.E.	Wald	df	Sig.	Exp(B)	95% C.I. for EXP(B)	
							Lower	Upper
**Male**								
Age (years)	−0.04	0.03	2.85	1	0.09	0.96	0.91	1.01
SBP (mmHg)	−0.00	0.01	0.07	1	0.80	0.99	0.97	1.02
Log Hs-CRP (mg/L)	−0.52	0.70	0.55	1	0.46	0.59	0.15	2.36
Ethnic specific WC	0.54	0.57	0.89	1	0.35	1.72	0.56	5.27
Log GGT	−0.04	0.67	0.00	1	0.96	0.97	0.26	3.55
Log PAL	0.12	1.51	0.01	1	0.94	1.12	0.06	21.47
Log CRP	0.09	0.08	1.29	1	0.26	1.10	0.94	1.28
Constant	1.99	2.41	0.68	1	0.41	7.33		
**Female**								
Age (years)	−0.06	0.03	3.40	1	0.07	0.94	0.88	1.00
SBP (mmHg)	−0.01	0.25	0.83	1	0.36	0.99	0.96	1.02
Log Hs-CRP (mg/L)	0.50	0.81	0.37	1	0.54	1.64	0.33	8.07
Ethnic specific WC	1.44	0.69	4.41	1	0.04	4.23	1.10	16.25
Log GGT	0.41	0.99	0.17	1	0.68	1.50	0.22	10.37
Log PAL	0.44	2.45	0.03	1	0.86	1.55	0.01	189.73
Log CRP	−0.03	0.09	0.13	1	0.72	0.97	0.82	1.15
Constant	3.53	3.18	1.24	1	0.27	34.24		

CRP = C-reactive protein; GGT_log = gamma glutamyl transferase log transformed; PAL = physical activity level; SBP = systolic blood pressure; WC = waist circumference ethnic specific cut point.

## Data Availability

The datasets used for analyses during the current study are not publicly available due ethical restrictions and participant confidentiality but are available from the corresponding author on reasonable request and in accordance to the NWU data sharing policy.

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
