# Peer review of "The Role of Physical Activity Status in the Relationship between Obesity and Carotid Intima-Media Thickness (CIMT) in Urban South African Teachers: The SABPA Study"

_ijerph, 2022, doi:10.3390/ijerph19106348_

Round 1
Reviewer 1 Report
The manuscript is very well written and moderately original, although it is of more interest to the study population than to the international community as it does not bring much novelty to what has already been described by the scientific community.
The study contemplates numerous variables and measurements to give robustness to its results and conclusions, which is appreciated, but in the face of so many results, it is recommended to include a figure that gathers its most significant results. Not a table, but a drawing or scheme would be sufficient as a graphical abstract. This would help to understand the results among so many measurements.
Author Response
The author thanks the reviewer for their constructive feedback.
Comments: The manuscript is very well written and moderately original, although it is of more interest to the study population than to the international community as it does not bring much novelty to what has already been described by the scientific community.
Response: Thank you for the positive comment.
Comments: The study contemplates numerous variables and measurements to give robustness to its results and conclusions, which is appreciated, but in the face of so many results, it is recommended to include a figure that gathers its most significant results. Not a table, but a drawing or scheme would be sufficient as a graphical abstract. This would help to understand the results among so many measurements.
Response: In our revised manuscript we included four graphs; and we could not do more than that because we had to comply with the IJERPH guidelines regarding the numbers of tables and figures in the manuscript.
Reviewer 2 Report
The manuscript (IJERPH-16669595) entitled “Low Physical Activity Status Curbed the Beneficial Impact of Physical Activity on the Central Obesity and Atherosclerosis Conundrum: the SABPA Study” focused on the association between PA, obesity, and CIMT in a multi-ethnic cohort of South African teachers.
It is surprising that there are so many overweight and obese people in South Africa. Also, it is surprising that most of them belong to the sedentary and light PA group. Nevertheless, the research results investigated in this study are not new. I'm not sure if the methods and results are scientifically correct. In addition, there is a need for further refinement in writing the manuscript.
Author Response
The author thanks the reviewer for their constructive feedback
Comments: The manuscript (IJERPH-16669595) entitled “Low Physical Activity Status Curbed the Beneficial Impact of Physical Activity on the Central Obesity and Atherosclerosis Conundrum: the SABPA Study” focused on the association between PA, obesity, and CIMT in a multi-ethnic cohort of South African teachers.
It is surprising that there are so many overweight and obese people in South Africa. Also, it is surprising that most of them belong to the sedentary and light PA group. Nevertheless, the research results investigated in this study are not new. I'm not sure if the methods and results are scientifically correct. In addition, there is a need for further refinement in writing the manuscript.
Response: We thank the reviewer’s positive comments. In terms of the observe high prevalence of overweight and obesity in the study, it is not strange because it is in line with the South Africa Demographic and Health Survey (SADHS) of 2016 in which it was revealed that 68% of women and 31% of men are overweight or obese as published by Statistics South Africa (STATSSA, 2016).
On the issue of the reviewer not sure if the methods and results are scientifically correct, we can confirm that indeed they are correct. We have refined our hypothesis and the study title to clarify your concern. The methods are based on a larger Sympathetic Activity and Ambulatory Blood Pressure in Africans as published by Malan et al., 2015. Further detail regarding the study protocol can be found, elsewhere. reference https://doi.org/10.1093/ije/dyu199.
Reviewer 3 Report
I would like to thank the Authors of the Manuscript “Low Physical Activity Status Curbed the Beneficial Impact of Physical Activity on the Central Obesity and Atherosclerosis Conundrum: The SABPA Study” for the opportunity to provide commentary on their work.
Overall, the Manuscript is well written, and every section is quite clearly presented, with the Discussion section being particularly rich in information. However, some corrections may be implemented to make the content of the article even stronger.
Firstly, and please take this as a matter of personal preference, I feel like the wording of the title may be too complex and definitive: the effect of low physical activity is given as a certainty, while all along the text it is given as a possibility. Moreover, the Authors report in their concluding remarks that “In line with the study hypothesis, significant positive relationships between CIMT and WC were observed in this study; however, including PA in the analysis did not yield any significant observable effect on the relationships between CIMT and body composition variables” (lines 388-391), so it is not clear to me why the title should give such relevance to the effect of low physical activity, when the study was not actually revelatory on that specific aspect.
Could the Authors justify the exclusion criteria that have been used to generate the study cohort? For example, how does lactation, psychotropic substance use or blood donation in the previous three months impact the acquisition/ quality of the data in this study (I presume it has something to do with blood pressure, heart rate and biochemical analysis)? I feel like this could be important information to find in the text, for a non-expert audience.
Moreover, I am wondering why a cohort of physically active teachers (for example, PA instructors) was not used as an extra control to test the beneficial effect of PA.
Does menopausal status have any impact on the physiological characteristics of the female participants in this cohort (or can you hypothesize an effect)? Recent studies seem to suggest that, at least in the case of obese women, there is a significant difference in CIMT between pre- and post-menopausal women, and menopause has been pointed out as a non-modifiable risk factor for increased CIMT (van Mil SR, Biter LU, van de Geijn GJM, et al. The effect of sex and menopause on carotid intima-media thickness and pulse wave velocity in morbid obesity. Eur J Clin Invest. 2019;49(7):e13118. doi:10.1111/eci.13118; Rychter AM, Naskręt D, Zawada A, Ratajczak AE, Dobrowolska A, Krela-Kaźmierczak I. What Can We Change in Diet and Behaviour in Order to Decrease Carotid Intima-Media Thickness in Patients with Obesity?. J Pers Med. 2021;11(6):505. Published 2021 Jun 3. doi:10.3390/jpm11060505; Łoboz-Rudnicka M, Jaroch J, Bociąga Z, et al. Impact of cardiovascular risk factors on carotid intima-media thickness: sex differences. Clin Interv Aging. 2016;11:721-731. Published 2016 May 23. doi:10.2147/CIA.S103521).
In the results section (lines 220-221) the Authors state that “Male teachers had a significantly higher prevalence of overweight (42%) compared to female teachers (27%)” but they do not provide a value of significance.
Table 4: some of the columns are cut at the edges and numbers disappear.
Along the text: sometimes the study is reported as “SABPA-study” and other times as “SABPA study” (without the hyphen).
Could you observe any difference related to ethnicity/ancestry in your cohort? For example, a recently published study on CIMT in populations of Central and Southern Africa highlighted genetic contributions to CIMT that were different than what already observed in populations of European ancestry (Boua PR, Brandenburg JT, Choudhury A, et al. Genetic associations with carotid intima-media thickness link to atherosclerosis with sex-specific effects in sub-Saharan Africans. Nat Commun. 2022;13(1):855. Published 2022 Feb 14. doi:10.1038/s41467-022-28276-x). I wonder if this may also be partly a reason why the present study and the one by Jin and colleagues provide different results.
Author Response
We thank the reviewer for their constructive feedback
Comments: I would like to thank the Authors of the Manuscript “Low Physical Activity Status Curbed the Beneficial Impact of Physical Activity on the Central Obesity and Atherosclerosis Conundrum: The SABPA Study” for the opportunity to provide commentary on their work.
Overall, the Manuscript is well written, and every section is quite clearly presented, with the Discussion section being particularly rich in information. However, some corrections may be implemented to make the content of the article even stronger.
Response: Thank you for the positive comments.
Comments: Firstly, and please take this as a matter of personal preference, I feel like the wording of the title may be too complex and definitive: the effect of low physical activity is given as a certainty, while all along the text it is given as a possibility. Moreover, the Authors report in their concluding remarks that “In line with the study hypothesis, significant positive relationships between CIMT and WC were observed in this study; however, including PA in the analysis did not yield any significant observable effect on the relationships between CIMT and body composition variables” (lines 388-391), so it is not clear to me why the title should give such relevance to the effect of low physical activity, when the study was not actually revelatory on that specific aspect.
Response: We take note of the complexity brought by the title and refinements are made on the title, notwithstanding the hypothesis and conclusion since it has a bearing on the title of the paper. The title has been revised to “The role of physical activity status in the relationship between obesity and carotid intima-media thickness (CIMT) in urban South African teachers: the SABPA study”
Comments: Could the Authors justify the exclusion criteria that have been used to generate the study cohort? For example, how does lactation, psychotropic substance use or blood donation in the previous three months impact the acquisition/ quality of the data in this study (I presume it has something to do with blood pressure, heart rate and biochemical analysis)? I feel like this could be important information to find in the text, for a non-expert audience.
Moreover, I am wondering why a cohort of physically active teachers (for example, PA instructors) was not used as an extra control to test the beneficial effect of PA.
Response: Your valuable observation is noted, as such in our revised manuscript we clarified your concern by addition of the statement ‘The exclusion followed in our study are grounded on the endocrine pathways associated with lactation which is ought to be linked to factors that influence systolic blood pressure such as arterial stiffness and compliance [O’Rourker et al., 1990; Malamo et al., 2016].’ The main aim of the larger SABPA study as published elsewhere (Malan et al., 2005), was epidemiological observational study not experimental in nature whereby a room to include the control group was possible. We are of the view that your observation could be something to be taken care of by future studies.
Comments: Does menopausal status have any impact on the physiological characteristics of the female participants in this cohort (or can you hypothesize an effect)? Recent studies seem to suggest that, at least in the case of obese women, there is a significant difference in CIMT between pre- and post-menopausal women, and menopause has been pointed out as a non-modifiable risk factor for increased CIMT (van Mil SR, Biter LU, van de Geijn GJM, et al. The effect of sex and menopause on carotid intima-media thickness and pulse wave velocity in morbid obesity. Eur J Clin Invest. 2019;49(7):e13118. doi:10.1111/eci.13118; Rychter AM, Naskręt D, Zawada A, Ratajczak AE, Dobrowolska A, Krela-Kaźmierczak I. What Can We Change in Diet and Behaviour in Order to Decrease Carotid Intima-Media Thickness in Patients with Obesity?. J Pers Med. 2021;11(6):505. Published 2021 Jun 3. doi:10.3390/jpm11060505; Łoboz-Rudnicka M, Jaroch J, Bociąga Z, et al. Impact of cardiovascular risk factors on carotid intima-media thickness: sex differences. Clin Interv Aging. 2016;11:721-731. Published 2016 May 23. doi:10.2147/CIA.S103521).
Response: We noted your comments, and in our revised manuscript we included menopausal status as a limitation by adding this statement ‘Additional limitation of the study was the lack of menopausal status data are given that some studies established that in obese women, significant differences in CIMT between pre-and post-menopausal women, and menopause exist [Łoboz-Rudnicka et al., 2016; van Mil et al., 2019; Rychter et al., 2021]’.
Comments: In the results section (lines 220-221) the Authors state that “Male teachers had a significantly higher prevalence of overweight (42%) compared to female teachers (27%)” but they do not provide a value of significance.
Response: We take note of the comment, and we deleted the word ‘significant’ in the sentence.
Comment: Table 4: some of the columns are cut at the edges and numbers disappear.
Response: In our revised manuscript amendments are made.
Comment: Along the text: sometimes the study is reported as “SABPA-study” and other times as “SABPA study” (without the hyphen).
Response: In our revised manuscript we change all the acronyms to SABPA study without the hyphen.
Comments: Could you observe any difference related to ethnicity/ancestry in your cohort? For example, a recently published study on CIMT in populations of Central and Southern Africa highlighted genetic contributions to CIMT that were different than what already observed in populations of European ancestry (Boua PR, Brandenburg JT, Choudhury A, et al. Genetic associations with carotid intima-media thickness link to atherosclerosis with sex-specific effects in sub-Saharan Africans. Nat Commun. 2022;13(1):855. Published 2022 Feb 14. doi:10.1038/s41467-022-28276-x). I wonder if this may also be partly a reason why the present study and the one by Jin and colleagues provide different results.
Response: We take note of your comments, and we note this as a limitation of our study and such a limitation has been added in our revised manuscript for consideration in that future study.
Round 2
Reviewer 2 Report
Thanks for your efforts.
Reviewer 3 Report
I would like to thank the Authors of this manuscript for the opportunity to assess the revised version of their work.
Overall, I am very pleased that my comments and criticisms have been properly addressed in the response to the reviewer and have been incorporated in the main text.
I find the manuscript sufficiently improved and its content far more clear and understandable than before. I have no ulterior comments to address.